# Engineering tRNA abundances for synthetic cellular systems

Akshay J. Maheshwari ●[1,3], Jonathan Calles ●[1,3], Sean K. Waterton[2] &
Drew Endy ●[1] ✉

Routinizing the engineering of synthetic cells requires specifying beforehand how many of each molecule are needed. Physics-based tools for estimating desired molecular abundances in whole-cell synthetic biology are missing. Here, we use a colloidal dynamics simulator to make predictions for how tRNA abundances impact protein synthesis rates. We use rational design and direct RNA synthesis to make 21 synthetic tRNA surrogates from scratch. We use evolutionary algorithms within a computer aided design framework to engineer translation systems predicted to work faster or slower depending on tRNA abundance differences. We build and test the so-specified synthetic systems and find qualitative agreement between expected and observed systems. First principles modeling combined with bottom-up experiments can help molecular-to-cellular scale synthetic biology realize design-build-work frameworks that transcend tinker-and-test.

Cellular-scale synthetic biology has not yet been made routine by any method. The practical impacts of synthetic genomics[1–6] and modeling cellular behavior[7–10] remain limited by scientific mysteries and technical complexities[11,12]. Functional abstractions well-developed for individual parts, devices, and pathways[13–16] often fail when combined within complex self-mixing molecular milieus comprising entire cells[17,18]. The history of routinization of engineering practice (e.g., structural, aeronautical, computational) suggests theory, modeling, and empiricism must be combined to progress further[19]. We hypothesized that a first principles approach for representing, designing, and building cellular-scale systems might enable functional abstraction appropriate to the design of cellular-scale systems and support future routinization.

Protein synthesis Using Recombinant Elements (PURE) is a defined system of 108 molecules, capable of expressing DNA. PURE is widely used as a starting point for bottom-up synthetic cell research[20,21]. PURE itself has been optimized via design and empirical testing[22–24]. Significant progress is being made toward engineering self-regenerating PURE[25,26]. We choose PURE as a suitable empirical system for testing computational frameworks that might improve routinization of cellular-scale systems engineering.

Colloidal Dynamics (CD) modeling enables unfitted simulation of cellular-scale system behavior with single molecule resolution[27]. Starting from molecular abundances and volume fractions we can compute expected cellular system behaviors[28]. We wondered if CD modeling could help design PURE-based systems that behave as expected.

Most latency in protein synthesis is due to physical transport of ternary complexes[28,29]. So, we focused on varying tRNA abundances in PURE. We created a Computer-Aided Design (CAD) tool, Colloidal Dynamics CAD (CD-CAD), to optimize tRNA abundance distributions for faster and slower protein synthesis. We applied CD-CAD to design optimized tRNA abundance distributions for synthetic cells designed for evolutionary containment[30].

We developed a method for building PURE systems that contain designer tRNA abundance distributions assembled directly via RNA synthesis, or Tunable Implementation of Nucleic Acids (TINA), to test our predictions. We used TINA to build synthetic cell-scale systems with CD-CAD specified tRNA distributions. We found that TINA worked well, qualitatively matching the performance specifications requested of our CD-CAD tool. CD-CAD plus TINA serves as an example of routinization of engineering for molecular-to-cellular systems via workflows that enable design-build-work outcomes.

[1]Department of Bioengineering, Stanford University, Stanford, CA 94305, USA. [2]Department of Biology, Stanford University, Stanford, CA 94305, USA. [3]These authors contributed equally: Akshay J. Maheshwari, Jonathan Calles. ✉e-mail: endy@stanford.edu

## Results

### Establishing an expected dynamic range for protein synthesis rates from tRNA abundances

We examined how variation from wild-type *E. coli* tRNA abundances impacts protein synthesis rates. We leveraged a colloidal-scale model for representing protein synthesis within cytoplasm[28]. Spontaneous diffusion of and interactions among individual ternary complexes, ribosomes, and native proteins are represented with nanometer and nanosecond resolution. The model enables first-principles estimation of translation elongation latencies (i.e., the time required to form peptide bonds) by directly simulating how long individual tRNA take to find and react with matching ribosomes. We developed a method for post hoc modification of relative tRNA abundances in prior simulations[28] to efficiently estimate how elongation latencies might change with changing tRNA abundances ("Methods").

We compared the expected performance of the wild-type *E. coli* tRNA distribution to a set of 10,000 randomly generated tRNA abundance distributions. To ensure all random distributions were non-trivial (i.e., remain capable of translating a cell-scale transcriptome), we required all tRNA maintain an abundance between the lowest (i.e., no trivially absent tRNA) and highest wild-type tRNA abundances (0.15% and 8.5% of total, respectively). We also held the total tRNA concentration fixed at ~225 μM[28]. Wild-type *E. coli* tRNA abundances are positively correlated with transcriptome-wide codon usage, unlike random distributions (Figs. 1A and S1 and S2), as expected[31,32]. The average elongation latency across the wild-type transcriptome was 193 ± 5.5 ms (mean ± SD) for wild-type tRNA abundances. Randomly generated tRNA abundance distributions were all expected to be slower in translating the wild-type transcriptome, with a mean expected elongation latency of 214 ± 5 ms (Fig. 1B).

### Rational design of tRNA abundance distributions to broaden dynamic range

We explored how rationally designed tRNA distributions might enable faster or slower average protein synthesis rates. We first explored a uniform distribution in which each tRNA is present at equal abundance, as previously established experimentally[30]. We found the correlation between transcriptome codon usage and matching tRNA was low for the uniform tRNA distribution (slope = 0.1) (Fig. 2A). The slight positive correlation arises from the degeneracy of coding; while most codons have a single matching tRNA the most frequent codon (CUG) happens to have two matching tRNA (Leu1 and Leu3). We simulated the expected elongation latencies across all *E. coli* transcripts, estimating an elongation latency of 214 ± 1.4 ms.

We developed stepwise correlated and anticorrelated tRNA distributions wherein each of the 40 unique tRNA are sorted in decreasing order of how often their corresponding codons appear in the transcriptome, and assigned stepwise decreasing or increasing relative abundances, respectively (Fig. 2B, C). tRNA corresponding to the most-used codons were assigned the highest relative abundance while tRNA corresponding to the least-commonly used codons were assigned the lowest relative abundance, or vice versa (lowest relative tRNA abundance was set to 0.12%, second-lowest was set to 0.24%, and so on, with highest relative abundance set to 4.8%, as needed to keep constant total tRNA mass). We found expected elongation latencies of 194 ± 5.0 ms (correlated) and 232 ± 3.8 ms (anticorrelated), the latter of which is ~20% slower than wild type.

We wondered if any tRNA abundance distribution might result in faster than wild-type elongation rates or if still-slower rates could be possible while still translating all transcripts. We engineered a codon-weighted tRNA distribution by exactly weighting tRNA to their corresponding codon usage (Fig. 2D). As anticipated, the codon-weighted distribution had a stronger positive correlation between transcriptome codon usage and matching tRNA (slope = 1.26) compared to both wild type and stepwise correlated. Predicted elongation latencies were 4% faster than wild type (185 ± 6.7 ms). We engineered a codon-weighted anticorrelated distribution that reversed the weighting, such that tRNA with the greatest corresponding codon usage were assigned the relative abundances of tRNA with the least corresponding codon usage and vice versa (Fig. 2E). As expected, the correlation between transcriptome codon usage and matching tRNA reversed, producing a stronger negative correlation (slope = −0.90); predicted elongation latencies were 23% slower than wild type (237 ± 4.2 ms).

### Computer-aided design of tRNA abundance distributions enables purposeful specification of protein synthesis rates

We developed a genetic algorithm (GA) to iteratively optimize a population of competing tRNA abundance distributions. The GA initialized via a population of random tRNA abundance distributions and then applied rounds of computational mutation and selection for better performing distributions ("Methods", Table S1). The performance of each individual tRNA abundance distribution was calculated using the colloidal dynamics simulator. The genetic algorithm plus underlying simulator result in a design tool that allows engineers to specify a desired translation elongation rate and receive a tRNA abundance distribution computed from first principles to produce the desired result. We named our tool Colloidal Dynamics Computer-Aided Design, or CD-CAD. We used CD-CAD to search for still-faster and still-slower tRNA abundance distributions, seeking to establish the

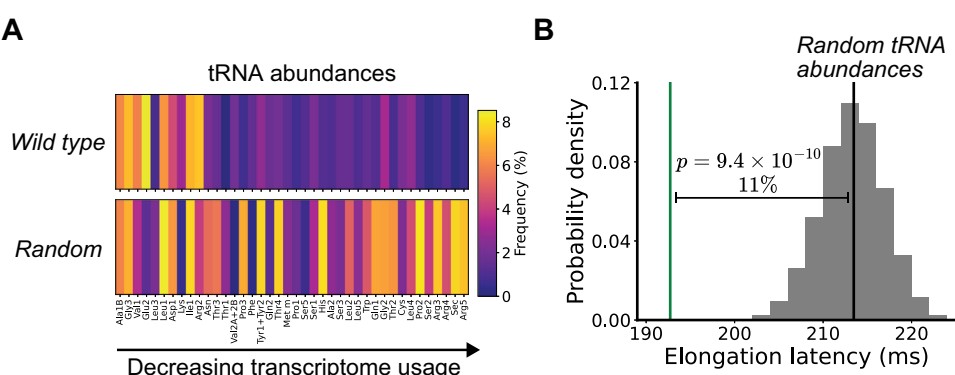

**Fig. 1 | Natural abundances of tRNA in wild-type *E. coli* should account for an 11% quickening of protein synthesis compared to random abundances. A** tRNA abundances (percentages, color bar) in wild-type *E. coli* compared to a representative random tRNA abundance distribution. tRNA are ordered based on decreasing frequency of cognate codons in the transcriptome. **B** Distribution for computed elongation latencies of randomly generated tRNA abundance distributions (gray; average: black line) compared with the elongation latency of the wild-type *E. coli* tRNA abundance distribution (green line). Source data are provided via a Zenodo repository (https://doi.org/10.5281/zenodo.7953836).

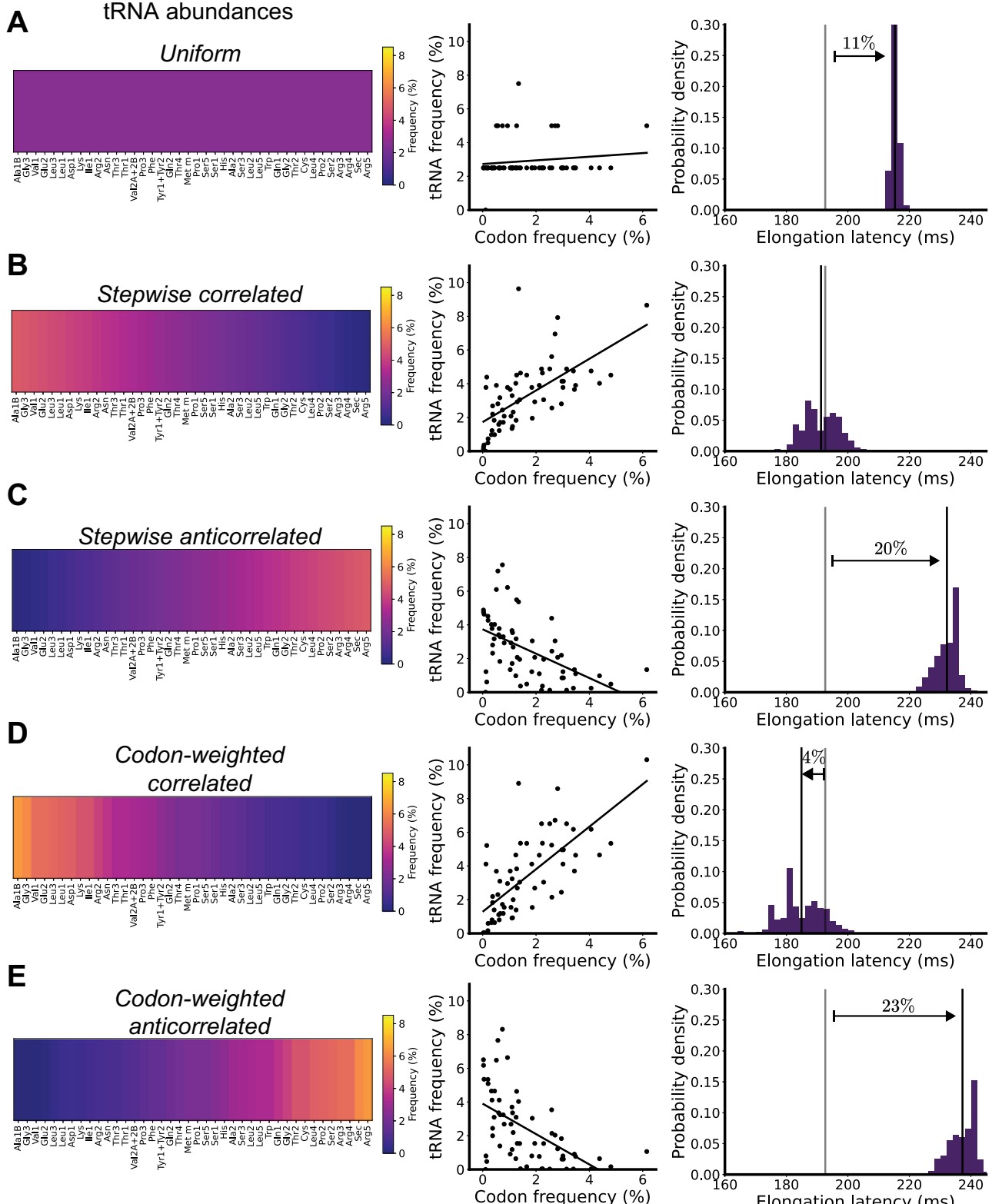

**Fig. 2 | Most rationally engineered tRNA abundance distributions should hinder protein synthesis.** For each tRNA abundance distribution (**A**–**E**): Left: relative tRNA abundances (percentages, color bar). tRNA are ordered based on decreasing frequency of cognate codons in the *E. coli* transcriptome. Middle: the fraction of cognate tRNA per codon (each codon is represented by its relative usage in the transcriptome). Right: the per-transcript elongation latency distribution (purple; average, black line) vs. that of wild-type *E. coli* (average, gray line; percentage change in elongation). Source data are provided via a Zenodo repository (https://doi.org/10.5281/zenodo.7953836).

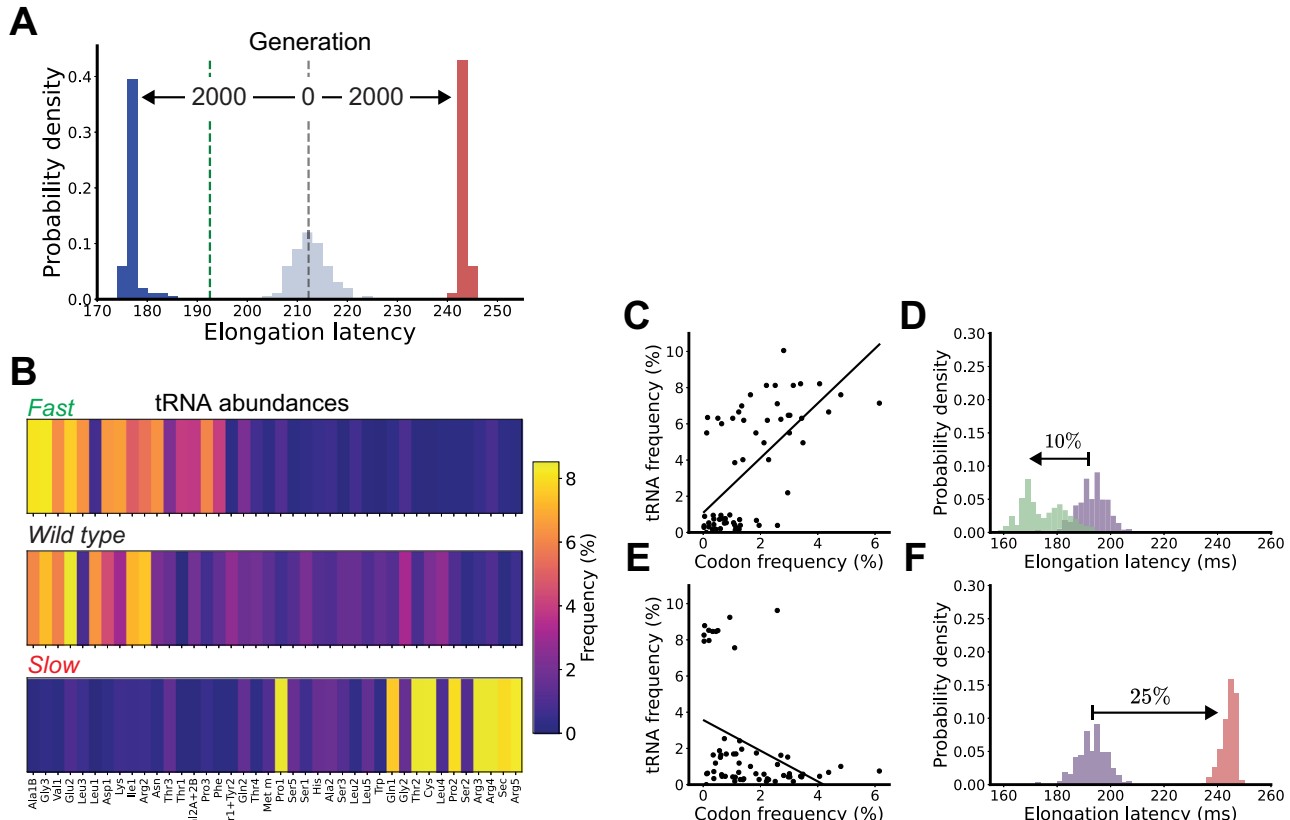

**Fig. 3 | Computer-aided design of tRNA abundances enables a broad dynamic range of specifiable translation elongation rates. A** Iterative selection of tRNA abundance distributions by CD-CAD enables either faster (blue) or slower (red) elongation compared to the initially random population that seeds the genetic algorithm (gray). Each distribution represents the expected average transcriptome-wide elongation latencies among individual tRNA distributions competing in the genetic algorithm (Methods); average elongation latencies computed with the wild-type tRNA distribution (green dashed line) and the uniform tRNA distribution (black dashed line). **B** tRNA abundance distributions produced by CD-CAD that

enable the fastest or slowest elongation compared to wild-type *E. coli* (color bar, percentages). **C** The fraction of cognate tRNA per codon (each codon represented by its relative usage in the transcriptome) for the tRNA distribution that enables the fastest elongation. **D** Per-transcript elongation latency distribution for the tRNA abundance distribution that enables the fastest elongation (green) vs. the wild-type abundance distribution (purple). **E, F** Same as (**C, D**) but for the tRNA abundance distribution produced by CD-CAD that enables the slowest elongation (red). Source data are provided via a Zenodo repository (https://doi.org/10.5281/zenodo.7953836).

full dynamic range of protein synthesis rates that might be obtained via precise engineering of tRNA abundances (Fig. 3A). To improve the likelihood that any designed tRNA distributions could be implemented for any transcript (i.e., organismal-scale functionality), we again constrained individual tRNA abundances to be within the bounds of naturally observed tRNA abundances and kept total tRNA abundance constant.

We selected the resulting fastest and slowest tRNA abundance distributions for further analysis. The fast-performing tRNA distribution assigned, in general, even greater relative abundance to tRNA with highly-used codons and even lesser relative abundance to tRNA with rarely-used codons; likewise, the slow-performing tRNA distribution assigned, in general, the opposite (Fig. 3B). We found that the fastest performing CD-CAD tRNA distribution exhibited a strong positive correlation (slope = 1.51) between transcriptome codon usage and matching tRNA compared to all rationally designed distributions (Fig. 3C). As desired, the elongation rates resulting from the computationally evolved tRNA distribution showed a systematic 10% average speed up across all transcripts with an average elongation latency of 175 ± 8.6 ms (Fig. 3D). Our analysis of the slowest performing tRNA distribution showed analogous results: the slow distribution exhibited a strong negative tRNA-codon usage correlation relative to all rationally designed distributions and the resulting protein synthesis rates showed a systematic 25% average slowdown across all transcripts with an average elongation latency of 244 ± 2.9 ms (Fig. 3E, F).

We noted that neither of the CAD-produced distributions were monotonically decreasing or increasing, unlike the earlier stepwise and codon-weighted distributions (Fig. 2). For example, in the fast distribution, the leucine tRNA Leu1 abundance is lowered despite being associated with the most-used codon (CUG). Since leucine tRNA Leu3 encodes the same codon (CUG), but also an additional codon (CUA), one possibility is that CD-CAD identified that redistributing relative abundance away from Leu1 should be a more efficient scheme.

To test this hypothesis, we performed a pairwise perturbation analysis in which we modified the relative abundance of each nearest-neighbor pair of tRNA (ordered by codon frequency) such that the relative abundance of the more frequent tRNA was 10% higher than the less frequent tRNA, while maintaining total relative abundance across the pair. We found that perturbations toward increased monotonicity, and in particular redistribution between Leu1 and Leu3, led to slower elongation latencies (Fig. S9A). Stacking of pairwise perturbations together and iteratively to further increase the monotonicity of the fast distribution led to slightly slower elongation latencies, validating the non-monotonic CAD-produced distribution as optimal and highlighting that elongation speed is relatively robust to small changes in the monotonicity of the tRNA abundance distribution (Figs. S9B and S10).

Given that codon choice is known to be biased across different genes[33], we also tested whether variation in optimized tRNA distributions for transcriptome sub-populations could underlie the CAD-

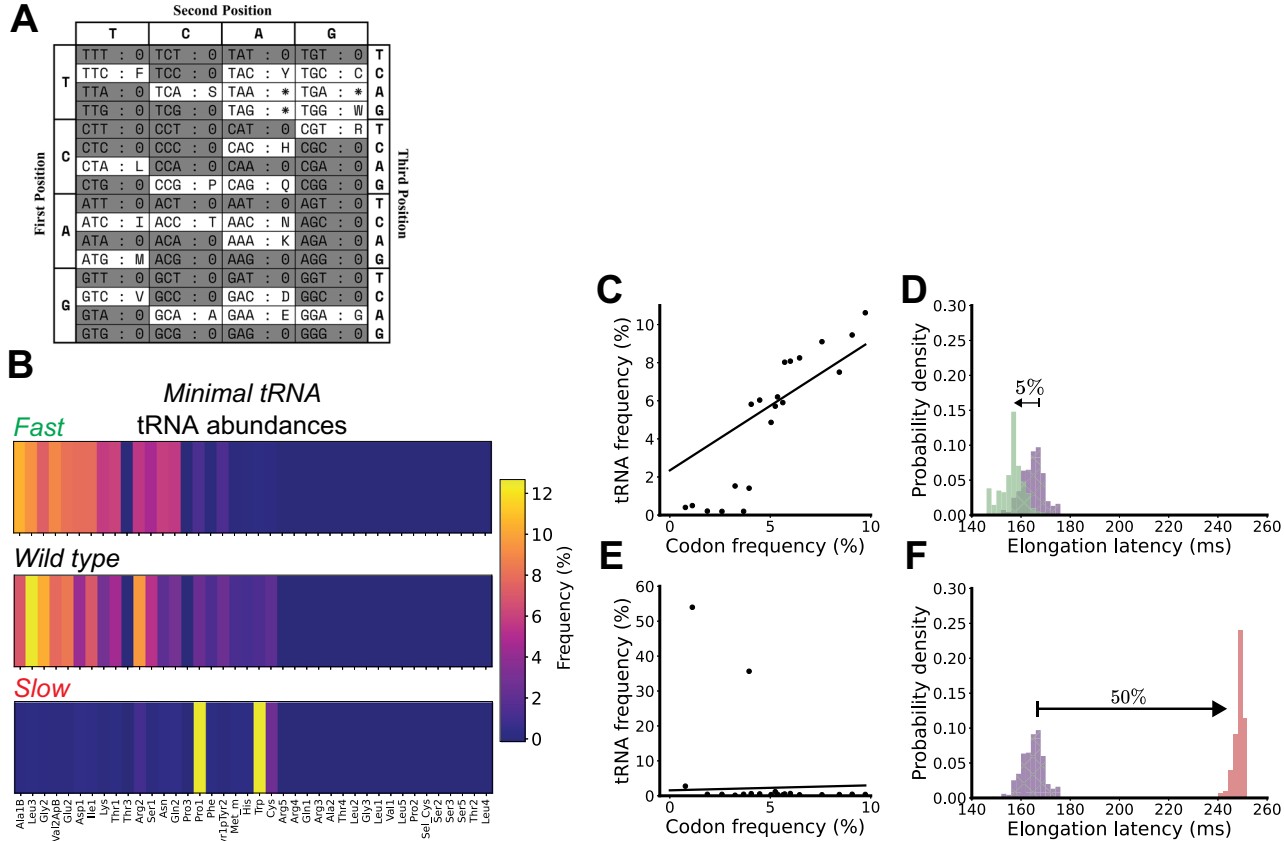

**Fig. 4 | Computer-aided design of tRNA abundances for quantitative control of synthetic organisms encoded by fail-safe genomes. A** Computationally designed fail-safe *E. coli* engineered to have only 20 sense codons (adapted from Calles et al.[30] under a Creative Commons CC BY license); white boxes represent sense codons. **B** Reduced tRNA abundance distributions produced by CD-CAD that enable the fastest elongation or slowest elongation compared to the wild-type distribution (color bar, percentages). **C** The fraction of cognate tRNA per codon, with each codon represented by its relative usage in the transcriptome, for the reduced tRNA abundance distribution that enables the fastest elongation. **D** Per-transcript elongation latency distribution for the reduced tRNA abundance distribution that enables the fastest elongation (green) vs. wild-type (purple). **E**, **F** are the same as (**C**, **D**) but for the reduced tRNA abundance distribution that enables the slowest elongation (red). Source data are provided via a Zenodo repository (https://doi.org/10.5281/zenodo.7953836).

produced non-monotonic distribution. As expected, we found that transcriptome sub-populations with high, intermediate, and low expression all had different optimal tRNA distributions (Fig. S11), indicating that codon-usage diversity across different gene sets is an important determinant of the optimal tRNA distribution for the full transcriptome. Notably, taking this stratification analysis to the limit and producing an optimal tRNA distribution for just a single gene with CD-CAD resulted in a non-monotonic distribution that could not be made faster by making the distribution more monotonic (Fig. S12). This finding supports the assertion that the degeneracy of translation itself is an important feature underlying CAD-discovered tRNA distributions, consistent with our findings in the pairwise perturbation analysis above.

### Designing tRNA abundances for quantitative control of synthetic cellular systems with fail-safe genomes

We previously reported designs for genetic codes that recognize only 20 sense codons, in which mutations in protein-coding sequences should most typically result in ribosome stalling and be selected against, regardless of the identity of the so-encoded proteins[30]. We refer to such codes as fail-safe codes because they are intended to constrain the evolutionary trajectories of so-encoded synthetic biological systems. We explored if CD-CAD could be used to design optimal tRNA abundance distributions for synthetic organisms encoded via fail-safe genomes.

We computationally recoded the *E. coli* transcriptome into one such fail-safe code, RED20, which only uses 20 sense codons (Figs. 4A and S3, "Methods"). tRNA abundances corresponding to null tRNA were reassigned to the tRNA encoding the same amino acid. We used our colloidal dynamics model to compute the elongation latencies of the so-modified transcriptome, estimating an expected average elongation latency of $164 \pm 4.6$ ms (-18% faster than wild-type *E. coli*).

We used CD-CAD to search for tRNA distributions capable of faster or slower elongation latencies (Fig. 4B). The fastest distribution had a positive correlation between matching tRNA and transcriptome codon usage (slope = 0.67) and an average elongation latency of $157 \pm 5.0$ ms (-5% faster than with wild-type abundances) (Fig. 4C, D). The slowest distribution was highly polarized, with nearly all tRNA abundance being assigned to tRNA Pro1 and Trp (two of the least common codons in the reduced-code transcriptome), corresponding to a slightly positive cognate tRNA-codon usage correlation (slope -0.14) and an average elongation latency of $248 \pm 1.9$ ms (-50% slower than wild-type abundances) (Fig. 4E, F).

### Design, construction, and experimental tests of CD-CAD engineered tRNA abundances for a synthetic cellular-scale system expressing a fail-safe encoded gene

We wanted to directly test whether designs produced via CD-CAD work as expected. We previously showed how PURE lacking all tRNA (PUREΔtRNA) can be supplemented with 21 synthetic tRNA

(20 elongator tRNA plus an initiator tRNA) at equimolar concentrations[30]. Here, we developed Tunable Implementation of Nucleic Acids (TINA) to directly construct any specified distribution of tRNA abundances (elongator tRNA plus initiator tRNA) from sequence-specified synthetic tRNA. Resulting tRNA sets can be combined with PUREΔtRNA to create functioning in vitro expression systems with custom tRNA abundance distributions ("Methods"). We used TINA to physically compose tRNA in precise relative ratios as specified by CD-CAD and measured protein expression using a RED20-encoded green fluorescent protein (GFP) reporter (Fig. 5A).

Using CD-CAD we designed two distributions, synFast and syn-Slow, to translate RED20encoded GFP faster and slower, respectively (Fig. 5B and Table S3). SynFast and synSlow elongation latencies were predicted to be 160 and 221 ms, respectively (i.e., synSlow is expected to translate GFP at 73% the rate of synFast). We used TINA to compose and test synFast and synSlow distributions in three independent batches (Fig. 5C–K). We numerically computed the derivatives of observed protein expression traces and extracted the average rate during the period of highest protein synthesis ("Methods"); we expected that elongation latency dominates translation kinetics during the period of maximum protein synthesis, after tRNA charging is at equilibrium and before PURE component depletion becomes rate limiting. We found that synFast and synSlow behaved as expected, with synFast demonstrating faster protein synthesis across all batches ($p = 0.30$ for batch 1, $p = 0.014$ for batch 2, and $p = 0.040$ for batch 3, using one-sided Kolmogorov–Smirnov tests) (Fig. 5E, H, K). We normalized translation rates for tRNA distributions within each batch by the average rate of synFast rates in the same batch. Combining results from all three batches, we found that synSlow translates slower than synFast ($66\% \pm 30\%$ vs. $100\% \pm 33\%$, respectively; $p = 0.0001$ using a one-sided Kolmogorov–Smirnov test), qualitatively matching our initial design specifications (Fig. S8). In our final batch of tRNA, we also prepared a uniform distribution of synthetic tRNA to directly compare synFast and synSlow to prior work[30]. Using our modeling we predicted that the uniform distribution would have an elongation latency of 177 ms (80% of SynFast). We found that uniform was slower than synFast, as expected ($62\% \pm 6.1\%$, respectively; $p = 0.004$ using a one-sided Kolmogorov–Smirnov test); there was no significant difference between uniform and synSlow ($62\% \pm 6.1\%$ vs. $74\% \pm 17\%$, respectively; $p = 0.35$ using a two-sided Kolmogorov–Smirnov test).

## Discussion

The engineering of synthetic cell-scale systems is not yet routine. We establish how molecular-to-cellular modeling can contribute to routinization of bottom-up engineering for a single cellular-scale process, protein synthesis. We developed Colloidal Dynamics Computer-Aided Design (CD-CAD) by overlaying a genetic algorithm on an unfitted first-principles colloidal dynamics model, enabling design of tRNA abundance distributions producing specified translation elongation rates. We also established Tunable Implementation of Nucleic Acids (TINA) to enable direct construction of any specified tRNA abundance distribution directly from synthetic RNA. Using both CD-CAD and TINA, we designed, built, and tested reduced tRNA distributions for an engineered reduced-codon synthetic system. Our TINA-implemented designs behaved as expected, qualitatively matching the performance specifications requested of CD-CAD.

We used CD-CAD to minimize or maximize one performance metric (i.e., elongation rate) of one cell system (i.e., translation). We did so as a proof of principle for using CAD tools to specify the behavior of cell systems generally. Our application of CD-CAD suggests that unfitted first-principles models can be used to direct design of molecular-to-cellular systems. Our tool did not require feedback or parameterization from our experiments, maintaining generalizability for translation system design across different transcriptomes. While the underlying colloidal dynamics simulation framework is generalizable, the model we used here is limited to translation elongation. Further models of cellular processes along with different optimization methods can be added to our CD-CAD framework to support design of other cellular-scale processes. We anticipate future practitioners will use such tools to specify performance in more nuanced ways. For instance, CD-CAD could be used to optimize translation rate to facilitate optimal co-translational folding of a target protein[34].

Our CD-CAD tool builds upon prior simulations[28] to enable low-cost design work. The computational cost of the colloidal physics simulations underlying our model was high due to the spatiotemporal resolution required for accurate molecular-resolution modeling coupled to the long simulation times needed to capture cellular-scale processes (e.g., up to ~300,000 CPU-hours costing ~$10,000). Here, we implemented a process for substituting tRNA distributions post hoc into already existing simulation data, effectively treating prior work as a training set to enable CAD work that is ~100,000-fold more computationally efficient (~1.5 h per optimization on a personal computer) ("Methods"). Constructing multipurpose simulation datasets with high up-front costs but broad generalizability is a widely used strategy that may be helpful for affordable routinization of molecular-to-cellular-scale design (e.g., pre-computed models in machine learning tools like AlphaFold[35]).

We constrained our initial use of CD-CAD to a total fixed tRNA concentration for three practical reasons. First, we wanted to focus on understanding, in isolation, how manipulating relative tRNA abundances might impact translation rates. Second, we expect that in building synthetic cellular systems, resource management will be key: shifting relative abundances among tRNA species, while maintaining fixed total resources, offers a resource-independent means for optimizing cell performance. Finally, by keeping tRNA concentrations fixed we could take advantage of prior colloidal physics simulations (above).

Our application of TINA, one-step precision integration of 21 RNA without requiring iteration, was enabled by our use of unmodified synthetic RNA as tRNA surrogates. In contrast to traditional extraction-based methods, direct synthesis allowed us to have full control over concentrations and fully avoid cross-species contamination. TINA relies on direct RNA synthesis[36,37] via commercial services, which are relatively expensive (e.g., ~$200,000 per μmol, list price) and slow (e.g., ~10 weeks from order to delivery). Moreover, synthesized RNAs can degrade over time (e.g., expression dropped an order of magnitude within one freeze-thaw cycle (Fig. 5C, F). We thus limited our experiments to a single reporter gene and were not here able to test if CD-CAD can account for sequence specific factors known to impact elongation rates (e.g., codon pair and dinucleotide biases[38]). Advances in RNA synthesis methods, including precision modification of RNA, will enable many exciting experiments and help make routine whole-cell synthesis.

Reduced genomes and tRNA sets should support faster protein synthesis. For example, we predicted that transcriptome-averaged elongation latency is 23% faster for a codon-reduced genome with a minimal tRNA set compared to a wild-type genome with a full tRNA set (164 vs. 214 ms, respectively). Our prediction makes sense at the limit: if a genome could be encoded by just one tRNA, protein synthesis should be incredibly rapid since mismatching tRNA would no longer need to be sampled and rejected by elongating ribosomes. While the wild-type tRNA distribution itself already prioritizes minimizing the split of tRNA abundance across isoacceptor tRNA (e.g., Leu1 has much greater relative abundance than Leu2, Leu3, Leu4, or Leu5), our model-generated distributions take this concept further, significantly reducing the split (Figs. 3 and S9 and S10).

Why are CAD-optimized tRNA distributions non-monotonic? Intuition suggests that tRNA abundances directly weighted by corresponding codon-frequency should lead to the fastest elongation rates.

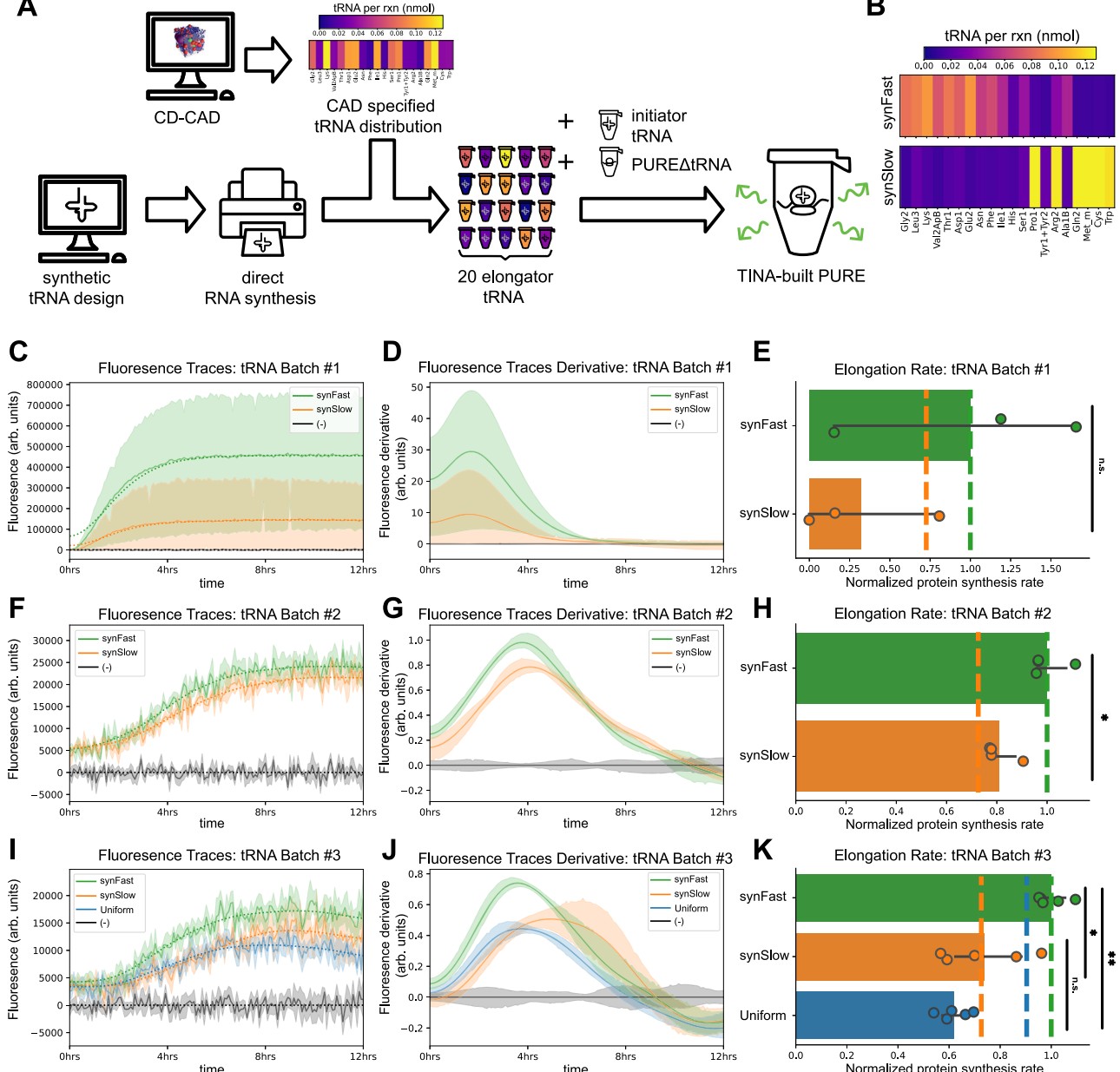

**Fig. 5 | Experimental construction and validation of CAD-engineered tRNA abundances in a synthetic system expressing a fail-safe encoded gene.** **A** Schematic overview of TINA with CD-CAD. We designed synthetic tRNA sequences that we sourced via direct RNA synthesis. We used CD-CAD to design tRNA abundance distributions and combined our synthetic tRNA as specified. We built in vitro translation systems by supplementing PUREΔtRNA with synthetic tRNA distributions plus synthetic initiator tRNA. Colors and color bars represent concentration of each tRNA species per reaction mixture. Icons adapted with permission from The Noun Project under a Creative Commons CC BY license. **B** CD-CAD specified tRNA distributions optimized for faster (synFast, above) and slower (synSlow, below) translation of RED20-encoded GFP. Color bars represent concentration of each tRNA species per reaction mixture. (**C**–**K**) Experimental data from tRNA batch #1 (**C**–**E**, *n* = 3 technical replicates), tRNA batch #2 (**F**–**H**, *n* = 4 technical replicates), and tRNA batch #3 (**I**–**K**, *n* = 5 technical replicates). Protein expression for three tRNA distributions is measured and analyzed: synFast (green), synSlow (orange), and uniform (blue, **I**–**K** only) along with a negative control,

PUREΔtRNA (black, labeled as (-)). Colors are used consistently across (**C**–**K**). Traces of GFP fluorescence over time (**C**, **F**, **I**), numerically computed derivatives of these traces (**D**, **G**, **J**) and calculated protein synthesis rates (**E**, **H**, **K**). In (**C**, **F**, **I**), solid lines represent the mean trace across replicates for each distribution. Shaded regions represent the 95% confidence interval in the estimate of the mean across replicates within each condition. Dashed lines represent the mean of traces individually smoothed with a Gaussian filter (see "Methods"). In (**D**, **G**, **J**), solid lines represent the mean of smoothed derivatives across replicates for each distribution. Shaded regions represent the 95% confidence interval in the estimate of the mean. In (**E**, **H**, **K**), error bars represent 95% confidence interval in the estimate of the mean and colors bars are mean values; colored dashed vertical lines are CD-CAD predicted rates. Solid black lines with labels show statistical significance (n.s. is not significant, *$p < 0.05$, **$p < 0.01$); one-sided Kolmogorov–Smirnov (KS) tests used for comparing synFast to all others, and a two-sided KS test used for comparing synSlow to Uniform. Source data are provided via a Zenodo repository (https://doi.org/10.5281/zenodo.7953836).

Our sensitivity analysis indicates that the phenomena of non-monotonicity in optimized tRNA distributions for natural transcriptomes arises from at least two underlying factors: (1) degeneracy of the translation code, enabling preferential choice between tRNA encoding similar codons during optimization (Figs. S9, S10 and S12), and (2) codon-usage variance across different subsets of the transcriptome leading to a variety of optimized distributions (Figs. S11 and S12). When these two phenomena are not present, such as in the optimization of a single gene in a non-degenerate translation code, CAD distributions and monotonic distributions perform similarly (Fig. S13). Compared to the codon-weighted tRNA distribution, we found that the CAD-optimized fast tRNA distribution was ~5% faster for the wild-type *E. coli* transcriptome, ~4% faster for the RED20 codon-reduced transcriptome, and ~2% faster for RED20-encoded GFP, highlighting that CAD optimization can enable performance gains especially-well in multi-gene systems with degenerate translation codes (Fig. S14).

Why are faster tRNA distributions not already adopted by wild-type *E. coli*? For example, given the direct relationship between elongation rate and growth rate[28,39] an 11% faster protein synthesis rate could seemingly improve fitness by 11%, providing a significant evolutionary advantage. Two explanations, among others, are: (1) accurate protein folding can be contingent upon fine-tuned protein synthesis rates, with too-fast protein synthesis leading to misfolded or aggregation-prone proteins[40]; (2) too-fast protein synthesis could also lead to excessive errors during translation, producing incorrect proteins that are less functional or saturate protein recycling systems. Such explanations may also explain our finding that elongation latencies are more broadly distributed for wild-type tRNA abundances compared to uniform abundances (Figs. 1 and 2). We anticipate that additional experiments testing functional protein production vs. overall protein production in bottom-up cell-building efforts using different engineered tRNA distributions will help resolve these possibilities and further strengthen model-based design capabilities.

Colloidal dynamics modeling and synthetic molecular composition can support molecular-to-cellular-scale synthetic biology. Routine engineering of synthetic cells will ultimately require design-build-work processes such as those described here for all life-essential processes. Integrated computer-aided design tools and system implementation methods supporting functional abstraction and reliable composition for transcription, metabolism, membrane biosynthesis and transport, replication, and cell division will be exciting to imagine and make real.

## Methods

### Estimating elongation latency for specified tRNA distributions

We used a previously established colloidal dynamics model of translation elongation dynamics and associated ensembles of thousands of translation voxel simulations to compute expected transcriptome elongation latencies[28]. Translation voxels are representative sub-volumes of cytoplasm that contain ternary complexes (aminoacylated tRNA bound to EF-Tu·GTP), ribosomes, and average-sized proteins representing all other native proteins; all molecules are approximated as spheres and represented at experimentally measured abundances. In translation voxel simulations, individual molecules move by Brownian motion with nanometer and nanosecond resolution and ternary complexes and ribosomes undergo reactions; reactions are modeled as a Markov process based on established intra-ribosomal kinetics with non-cognate, near-cognate, cognate reactions having different latencies and outcomes. Cognate reactions are represented as able to succeed or fail while non-cognate and near-cognate reactions are represented as always failing (i.e., misincorporations are not considered). The elongation latency of any single voxel is calculated as the time for a tracked ribosome loaded with a particular codon to successfully react with a cognate ternary complex. A representative elongation latency is derived via an ensemble average across thousands of simulated voxels that span the statistical variation of molecular abundances and transport and reaction dynamics.

We developed a method for post hoc modification of relative tRNA abundances in already existing voxel simulations to estimate elongation latencies for any specified transcriptome and tRNA abundance distribution. We noted that prior analysis already computed component elongation latencies for translation voxels across all possible combinations of relative cognate and non-cognate ternary complex abundances (with fixed total tRNA abundance) as a sub-process in computing overall elongation latency. We utilized this feature to compute elongation latency with any specified tRNA distribution and transcriptome as follows:

(1) Estimate the likelihood of different cognate:non-cognate ternary complex ratios in a translation voxel for each possible codon via Monte Carlo simulations that sample tRNA based on their specified relative abundances (simulations with zero cognates are approximated as having one cognate to avoid trivial distributions).

(2) Compute the elongation latency for each possible codon as a weighted sum of elongation latencies for each pre-computed cognate:non-cognate ratio component elongation latency, based on likelihoods computed in (1).

(3) For each transcript in a specified transcriptome, compute the relative abundance of each codon.

(4) Compute the elongation latency of each transcript in a specified transcriptome as a weighted sum of elongation latencies computed for each codon (2), based on relative likelihood of each codon computed in (3).

(5) Compute the overall transcriptome elongation latency by averaging the elongation latency of each transcript, weighted by the frequency of each transcript.

This analysis can be used generally to test the impact of specified tRNA distributions with different transcriptomes (e.g., *E. coli* transcriptome, $\mu = 1.0$ dbl/h used here), genetic codes (e.g., RED20 used here), and single transcripts (e.g., RED20-encoded GFP used here).

### Optimizing tRNA distributions using a genetic algorithm (CD-CAD)

Our Colloidal Dynamics Computer-Aided Design tool (CD-CAD) uses a genetic algorithm wrapped around the colloidal dynamics model. We initialized our genetic algorithm by generating a population of $N = 100$ random relative tRNA abundance distributions, $\mathbf{D}_1, \mathbf{D}_2, \ldots, \mathbf{D}_{100}$, each with 40 unique relative tRNA abundances, $\mathbf{D}_{i,1}, \mathbf{D}_{i,2}, \ldots, \mathbf{D}_{i,40}$, such that $\sum_{j=1}^{40} \mathbf{D}_{i,j} = 1$. We then iteratively executed the following for 2000 generations, realizing convergence in all our simulations:

(1) For each tRNA distribution in the population we computed and averaged the elongation latencies of the full specified transcriptome, $<\tau_{\text{elong},1}>, <\tau_{\text{elong},2}>, \ldots, <\tau_{\text{elong},100}>$.

(2) Depending on whether we were optimizing for faster- or slower-performing tRNA distributions, we computed the fitness of each tRNA distribution as follows:

$$\text{Fast}: \text{For } i = 1, 2, \ldots, 100, \text{Fitness}(\mathbf{D}_i) = \frac{<\tau_{\text{elong},i}>^{-1}}{\sum_{j=1}^{N} <\tau_{\text{elong},j}>^{-1}}, \quad (1)$$

$$\text{Slow}: \text{For } i = 1, 2, \ldots, 100, \text{Fitness}(\mathbf{D}_i) = \frac{<\tau_{\text{elong},i}>}{\sum_{j=1}^{N} <\tau_{\text{elong},j}>}. \quad (2)$$

(3) We then removed the 10 tRNA distributions with the lowest fitness from the population.

(4)  From the remaining tRNA distributions, we randomly sampled five pairs of unique tRNA distributions with choices for each distribution weighted by their fitness, p($\mathbf{D}_i$) = Fitness($\mathbf{D}_i$).

(5)  For each pair, ($\mathbf{D}_A$, $\mathbf{D}_B$), we swapped the values of eight randomly chosen relative tRNA abundances within each distribution using a recombination rate of 0.2.

(6)  For each pair, ($\mathbf{D}_A$, $\mathbf{D}_B$), we then used a mutation rate of 0.1 to randomly select four tRNA and changed their abundance values to a randomly chosen value between the minimum and maximum of wild-type relative tRNA abundances (0.15% and 8.52%, respectively, for the full transcriptome, or 1.2% and 12.8%, respectively, for the reduced-code transcriptome).

(7)  Finally, we added the 10 recombined and mutated tRNA distributions to the population.

Our algorithm was implemented with parallel processing using Python 3.7.4 and took ~1.5 h per tRNA distribution optimization routine on a personal laptop (2.3 GHz 8-Core Intel Core i9, 32 GB 2667 MHz DDR4).

### Developing the codon-reduced transcriptome

We computationally replaced each codon in the wild-type *E. coli* transcriptome (μ = 1.0 dbl/h) not represented in the minimal-code *E. coli* (Fig. 4A) with the codon represented in the minimal-code *E. coli* that encoded the same amino acid (e.g., all UUU were converted to UUC). We used the MG1655 genome with gene and sequence data from Ecocyc[41] and Caglar et al.[42].

### Preparing expression plasmids

We previously prepared a RED20-encoded GFP reporter plasmid (pSB1C3::pT7::sfGFP_RED20) and glycerol stocks of *E. coli* Top10 cells carrying this plasmid[30]. We plated cells on LB agar with chloramphenicol (25 ng μl⁻¹) and grew overnight at 37 °C to isolate individual colonies. We outgrew a single colony in 100 ml Terrific Broth (TB) with chloramphenicol (25 ng μl⁻¹) overnight at 37 °C with shaking. We split the overnight culture into 20 batches of 5 ml each and prepared each batch separately using QIAprep Spin Miniprep kits (QIAGEN, Cat No./ID: 27104) producing 20 × 50 μl preps. We pooled preps into 5 batches of 200 μl to purify and concentrate using QIAquick PCR & Gel Cleanup kits (QIAGEN, Cat No.ID: 28506). Final DNA product was assessed for quantity and purity using a NanoDrop 2000 (Thermo Scientific). An annotated sequence map for sfGFP_RED20 is freely available (https://benchling.com/s/seq-w63RBxrXRxi6uIruvKEM) and a physical copy has been deposited with Addgene (Plasmid #135173).

### Preparing specified tRNA distributions for in vitro protein expression assays (TINA)

We sourced 21 tRNA sequences individually by direct RNA synthesis without any base modifications (Agilent Technologies)[30] (Table S2). Each RNA was resuspended individually in nuclease free water to a putative concentration of 100 μM, as quantified by the Qubit RNA High Sensitivity (HS kit) (Thermo Scientific, Cat No. ID: Q32852) and stored at −80 °C.

Each tRNA distribution tested in this work used 200 μM total elongator tRNA, or one-fifth the concentration previously used (see Calles et al.[30]) to use concentrations closer to physiological values[31] (Fig. S4). For example, the uniform distribution had a final concentration of 10 uM of each of the 20 elongator tRNA. We added 10 μM initiator tRNA to each distribution because we found this concentration optimizes expression rates (Fig. S5). To prepare tRNA distributions for in vitro expression assays, we calculated the volume of each tRNA stock required then manually pipetted each tRNA into a fresh PCR tube to form a separate dilute tRNA stock for each condition. We dried each tRNA stock in a Vacufuge centrifuge concentrator

(Eppendorf) at 45 °C under vacuum, then resuspended in nuclease free water to produce 5x concentrated synthetic tRNA stock mixes ready for downstream use. We stored stock mixes at 80 °C until ready for use.

### Expressing protein and measuring fluorescence in vitro

For each in vitro expression reaction, we mixed the following reagents supplied in the PUREΔtRNA kit: 1 μl Solution A (minus aa, tRNA), 1.5 μl Solution B, and 0.5 μl amino acid master mix (3 μl total). To each mixture, we added 0.5 μl murine RNase inhibitor at 40 U μl⁻¹ (NEB# M0314S), 60 fmol of pSB1C3::pT7::sfGFP_RED20, and nuclease free water to 4 μl final volume, giving us PURE reaction master mixes. We made PURE expression reactions by combining 4 μl PURE reaction master mix and 1 μl of 5x synthetic tRNA stock mixes (preparation described above).

We performed reactions in black/clear bottom, 384-well microtiter plates (Corning). For any given experiment, we assembled all reaction mixtures for a given condition as a single master mix, then pipetted 5 μl per replicate into separate wells. We carried out reactions in a SpectraMax i3 plate reader (Molecular Devices) at 37 °C for more than 12 h, measuring protein expression by spectroscopy (excitation at 485 nm ± 9 nm; emission at 535 nm ± 15 nm) every 5 min.

### Analyzing data from in vitro expression assays

We used Python 3.10 and its scientific analysis and visualization ecosystem (numpy 1.23.5, scipy 1.9.3, pandas 1.5.2, matplotlib 3.6.2, seaborn 0.12.1)[43,44]. We first took individual fluorescence timeseries for each replicate and smoothed them with a Gaussian filter to remove high frequency noise. We then calculated background signal by averaging smoothed traces across replicates of PUREΔtRNA and subtracted this background signal from all smoothed traces. We numerically calculated the derivative of each trace and smoothed the derivative with a Gaussian filter. We sought to extract the protein production rate during the period of maximal protein production. To do so, we took the average of each smoothed derivative trace around the vicinity of the max value of the derivative (above 50% of max value both before and after the max) (Fig. S6). We show that our analysis is qualitatively robust to threshold (Fig. S7).

### Statistics and reproducibility

Sample sizes for simulations were determined based on replicates required to distinguish between conditions. For experiments, we conducted as many replicates as possible given how much of our limiting reagent we could source (synthetic tRNA). No data were excluded from the analyses. The experiments were not randomized. The investigators were not blinded to allocation during experiments and outcome assessment.

### Reporting summary

Further information on research design is available in the Nature Portfolio Reporting Summary linked to this article.

## Data availability

All input and output simulation files and all raw and processed experimental data are available via a Zenodo data repository: https://doi.org/10.5281/zenodo.7953836 (https://zenodo.org/record/7953836)[45]. Genome and sequence data from Ecocyc[41] and Caglar et al.[42] are also provided in the Zenodo repository.

## Code availability

All computer code is available on Github (https://github.com/EndyLab/tRNACAD) and has also been deposited on Zenodo: https://doi.org/10.5281/zenodo.8088058 (https://zenodo.org/record/8088058)[46].

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

## Acknowledgements

We thank Anton Jackson Smith for help with plate reader measurements, Jeff Sampson for help with synthetic RNA design, and Agilent for supplying synthetic RNA. Funding was provided by NIH R01GM086663 to D.E., NSF 1929752 to D.E., and NIH T32 Training Grant GM007365 to A.J.M.

## Author contributions

A.J.M., J.C., and D.E. conceptualized the project and drafted the manuscript. A.J.M. performed all modeling and simulations. J.C. and S.K.W. performed all experiments. All authors assisted with data analysis and reviewed and approved the final manuscript.

## Competing interests

A.J.M. and J.C. are founders of b.next, a synthetic biology company routinizing the engineering of cells. All other authors declare no competing interests.
