## [Peer Review File · Nature Communications]

Reviewers' Comments:

Reviewer #1:

Remarks to the Author:

In this paper the authors demonstrate the ability for a colloidal dynamics model to enable the prediction of translation dynamics of a gene to varying tRNA distributions. They demonstrate how different tRNA distributions can affect per-transcript elongation times, showing that translation is not optimized purely for speed. The model is then used to design PURE reactions with defined tRNA distributions to affect the translation time of a fluorescent reporter in a desired way. The results of these experiments showed qualitatively matching results to the models, although a large experiment to experiment variability.

The paper is clear, well-structured, and nicely written, although I found it perhaps a little too concise in places – some expansion of ideas and result would have improved readability. The figures on the whole were appropriate and summarized the key results well, but there formatting could be greatly improved (e.g., font size). It is difficult for me to comment on the accuracy of the computational aspects because the source code was not provided. However, the brief high-level descriptions of the methods used seemed appropriate and sound. I do believe though that well-documented code needs to be included with the publication (not provided on request) given the central role of the simulations and modelling approach in the work.

Overall, I commend the authors on a very interesting paper that will be of great interest to the synthetic biology and build-a-cell communities. However, my main concern is that I felt it lacked the scope/scale I would normally expect in a paper published by Nature Communications. In several places (highlighted below) results were not fully explored and the experiments are rather limited in their scope. For example, there are a lot of outstanding questions regarding the role of translational dynamics, codon optimization and tRNA availability that this system is primed to answer. While the authors may feel that exploring some of these are outside the scope of this work, I do feel it would really help demonstrate the value of this approach beyond a couple simple examples and help increase the impact of the work. I leave it to the Editor to decide whether such additional experiments are necessary, but I do feel they would further strengthen the message of the work, verify the approach more thoroughly, and bring in a wider readership to the piece.

My main comments that need addressing are below:

Page 2, Line 49: "worked, well matching" -> "worked well, matching"

Page 3, Line 54: While translation rate is correlated with tRNA abundance, the absolute translation time of a tRNA is not trivial to measure. Some details of this in the main text would be helpful given it is not obvious how this connection can be made.

Page 3, Line 80: What do you mean by a "broader range of protein synthesis rates"? Bigger differences from min to max? It is currently unclear.

Page 4, Line 90: "assigned stepwise decreasing or increasing relative abundances" – how were the absolute abundances calculated though to give times in absolute units? This is not clear at all?

Page 6, Line 131: You mention about the non-monotonic feature of the distributions found but say very little about why this might be. Some further investigation into the cause of this feature I feel is warranted. For example, given a codon is always in a context of the broader gene, it seems that correlations of codon types highly or lowly expressed genes could enable this type of feature to emerge. It has also been shown that synonymous codon choice within a gene is often highly biased (see: 10.1016/j.cell.2010.02.036), which may be another route to this outcome. Either way, I'd expect more effort to unpin the cause of this effect.

Page 9: No estimated elongation rates from the experiments are given. Why not? Would it not be possible to calculate this given the system is controlled to properly assess the accuracy of the model?

Page 9: A comparison of several different tRNA distributions is shown for a single fluorescent reporter. This is a rather limited experiment and would benefit from seeing whether similar predictive power is seen for other types of protein. For example, considering genes where biases in amino acids are seen would offer a chance to explore some of the questions that remain around the encoding of genes and the effects these have on translational dynamics.

Page 10: Why is there such large variability between batches? It seems like the expended trends are observed, but it is a little too strong to say they "matched" the design.

Page 11: I struggled to fully understand how you leveraged previous simulation data in this work. It would have been far more helpful to include the source code of the simulations and extended model to be able to assess precisely what has been done. Why was this not included?

Figure 1A: Scale bar is not labeled, and text is impossible to read on the x-axis. It would be helpful to ensure all text is of a similar and readable size (e.g., 8pt). This comment relates to all the figures, as text size in many of them is a problem that needs to be fixed.

Reviewer #2:

Remarks to the Author:

he manuscript addresses an interesting element of synthetic biology considering engineering of synthetic cells specifically the required abundance of different tRNAs. The manuscript reports on the generation of a genetic algorithm to do this. The manuscript is clearly written and combines the development of the computational model with in vitro validation studies.

I have the following comments:

1. The manuscript had two conditions on tRNA abundance, I understand the need for the first one, but I wonder if it could be explained why the overall tRNA concentration was constant.
2. In the first part we see the the tRNA abundance based on codon frequency performs better than wild type. The CD-CAD approach alter this and has higher frequencies of the most widely used tRNAs. How much faster (better) are these abundances proposed by CD-CAD than using the abundances that correlate with codon usage? i.e. what is the benefit of CD-CAD over this much simpler optimisation? Is the cost worth the improvement?
3. Figure 3 it would be good to avoid the combination of red and green in this figure.
4. Do we always want to optimise the rate of protein synthesis? Changes in protein folding, (although often through codon optimisation) have shown that they can affect the folding of proteins and alter their function? Is there any consideration for this? i.e. would it always be good to speed up the rate of translation? There is some thought on this in the discussion but it could be expanded.

Reviewer #1 (Remarks to the Author):

In this paper the authors demonstrate the ability for a colloidal dynamics model to enable the prediction of translation dynamics of a gene to varying tRNA distributions. They demonstrate how different tRNA distributions can affect per-transcript elongation times, showing that translation is not optimized purely for speed.

The model is then used to design PURE reactions with defined tRNA distributions to affect the translation time of a fluorescent reporter in a desired way. The results of these experiments showed qualitatively matching results to the models, although a large experiment to experiment variability.

The paper is clear, well-structured, and nicely written, although I found it perhaps a little too concise in places – some expansion of ideas and result would have improved readability. The figures on the whole were appropriate and summarized the key results well, but there formatting could be greatly improved (e.g., font size).

It is difficult for me to comment on the accuracy of the computational aspects because the source code was not provided.

However, the brief high-level descriptions of the methods used seemed appropriate and sound. I do believe though that well-documented code needs to be included with the publication (not provided on request) given the central role of the simulations and modelling approach in the work.

We thank the reviewer for their feedback and thorough reading of our manuscript. We apologize for not including a link to all code used in our modeling and simulations. A Github repository containing all code has been linked in the manuscript and here: <https://github.com/EndyLab/tRNACAD>

Overall, I commend the authors on a very interesting paper that will be of great interest to the synthetic biology and build-a-cell communities.

Thank you!

However, my main concern is that I felt it lacked the scope/scale I would normally expect in a paper published by Nature Communications. In several places (highlighted below) results were not fully explored and the experiments are rather limited in their scope.

For example, there are a lot of outstanding questions regarding the role of translational dynamics, codon optimization and tRNA availability that this system is primed to answer. While the authors may feel that exploring some of these are outside the scope of this work, I do feel it would really help demonstrate the value of this approach beyond a couple simple examples and help increase the impact of the work. I leave it to the Editor to decide whether such additional experiments are necessary, but I do feel they would further strengthen the

message of the work, verify the approach more thoroughly, and bring in a wider readership to the piece.

We thank the reviewer for their comment overall and suggestions about scope.

We have significantly expanded the scope of our theoretical and computational work, as described below, focusing on sensitivity analyses that reveal mechanisms underlying monotonic versus non-monotonic tRNA abundance distributions, and other aspects. These complementary results help explain and strengthen our primary findings. We feel additional experiments about the science of tRNA abundance distributions are well beyond the scope of the work reported here.

For example, we note our manuscript reports doing and then combining several novel ideas for the first time (i.e., tunable assembly of tRNA distributions from direct RNA synthesis; colloidal-scale molecular-to-cellular Computer Aided Design). Taken together, we are showcasing how such tools can enable rational quantitative engineering of cellular-scale systems. Doing so required operating at the limits of what is possible in terms of RNA synthesis, computer simulation, and experiment. The fact that many questions are now addressable in new ways for the first time is an exciting outcome of our work.

My main comments that need addressing are below:

Page 2, Line 49: “worked, well matching” -> “worked well, matching”

Thank you; corrected.

Page 3, Line 54: While translation rate is correlated with tRNA abundance, the absolute translation time of a tRNA is not trivial to measure. Some details of this in the main text would be helpful given it is not obvious how this connection can be made.

We apologize for the ambiguity. We have modified the main text and methods sections of the manuscript to explain fully. For reference, in prior work we built a colloidal physics model that simulates the chemistry and physics of individual molecules to estimate elongation latencies starting from only molecular identities, abundances, and cell volume as inputs [Maheshwari et al., 2022, mBio]. The model enables first-principles estimation of translation elongation latencies (i.e., absolute time required to form peptide bonds) by directly simulating how long individual tRNAs take to find and react with matching ribosomes. In this prior work we established an expansive set of thousands of simulations that span the statistical variation of molecular abundances and transport and reaction dynamics found in wild-type cytoplasm. In our work here, we developed a new method for post-hoc modification of relative tRNA abundances to estimate how elongation latencies might change given non-wild type tRNA abundances.

Page 3, Line 80: What do you mean by a “broader range of protein synthesis rates”? Bigger differences from min to max? It is currently unclear.

Thank you for pointing out the need for clarification. We meant faster or slower average protein synthesis rates and have clarified the text.

Page 4, Line 90: “assigned stepwise decreasing or increasing relative abundances” – how were the absolute abundances calculated though to give times in absolute units? This is not clear at all?

Thank you for pointing out that our description was not clear. We assigned stepwise increasing or decreasing relative abundances. I.e., the least common tRNA by corresponding codon abundance was set to a relative abundance of 0.12%, the second was set to 0.24%, and so on, with the highest abundance set to 4.8%. We then used these tRNA abundance parameters within our colloidal physics simulations to estimate an elongation latency, as described in Maheshwari et al., mBio (2022) in units of absolute time. We have expanded and clarified the main text and methods sections.

We also realized we mislabeled our figure references to Figure 2 which may have caused some confusion; we apologize for and have corrected this error.

Page 6, Line 131: You mention about the non-monotonic feature of the distributions found but say very little about why this might be. Some further investigation into the cause of this feature I feel is warranted. For example, given a codon is always in a context of the broader gene, it seems that correlations of codon types highly or lowly expressed genes could enable this type of feature to emerge. It has also been shown that synonymous codon choice within a gene is often highly biased (see: 10.1016/j.cell.2010.02.036), which may be another route to this outcome. Either way, I'd expect more effort to unpin the cause of this effect.

We thank the reviewer for their keen observation about the non-monotonicity of genetic-algorithm produced distributions. We have expanded our analysis and discussion of non-monotonicity as described below. Overall, we found that non-monotonicity is a function of two phenomena: (1) degeneracy of the translation code enabling preferential choice between tRNA encoding similar codons during optimization (Figure S9, S10, S12), and (2) codon-usage variance across different sets of genes leading to a variety of optimized distributions, as hypothesized by the reviewer (Figure S11, S12). When these two phenomena are not present, such as in the optimization of a single gene in a non-degenerate translation code, CAD-distributions and monotonic distributions perform similarly (Figure S13). We have added two new paragraphs in the Results, one paragraph in the Discussion, and six new Supplementary Figures addressing these points, described in more detail below.

We started our analysis of non-monotonicity with a sensitivity analysis of the fastest tRNA distribution found by CD-CAD for the *E. coli* transcriptome (Figure 3C and 3D). We performed a pairwise tRNA perturbation analysis in which we perturbed the relative abundance of each tRNA and its nearest codon-weighted-frequency neighbor to produce a more monotonic distribution (example of one such perturbation in Figure S9A, top). Specifically, for each tRNA (sorted by the frequency of their corresponding codons) we moved relative abundance to its immediate more codon-weighted-frequent neighbor such that the more frequent neighbor had a 10% higher relative abundance. For example, we perturbed

Val1 to be 10% higher than its neighbor Glu2 (Figure S9A, top). We then computed an average elongation latency from N=3000 runs of each perturbed distribution. We found that nearly all perturbations led to slower elongation latencies (Figure S9A, bottom), consistent with CD-CAD discovering a fastest distribution. Interestingly, perturbing the Leu1-Leu3 and Asp1-Leu1 pairs led to the largest slowdown in elongation latency (0.5% and 0.9% respectively). This finding is consistent with the conclusion that maintaining Leucine-encoding tRNA in only Leu1 as opposed to splitting abundance across Leu1 and Leu3 is a key strategy discovered by CD-CAD; a similar phenomenon is seen with Thr3 and Thr1 perturbations (0.35% and 0.15%), suggesting a general mechanism underlying non-monotonicity in CD-CAD-discovered distributions.

We further assessed the effect of multiple simultaneous perturbations towards more monotonic distributions (Figure S9B). We performed sequential pairwise tRNA perturbations, adding each additional perturbation to the already perturbed distribution (i.e., shifting the distribution one pair at a time towards an overall more monotonic distribution). The result was a distribution with 39 sequentially added perturbations. We found that each additional perturbation overall led to slower average elongation (Figure S9B bottom, N=3000 runs), consistent with what we would expect with additional deviation from the CD-CAD discovered optima. We took this analysis to the limit by performing the 39-sequential perturbation over multiple iterations until we produced a fully monotonic distribution (Figure S10A and S10B). We measured monotonicity by computing the percent of nearest-neighbor tRNA pairs that had more relative tRNA abundance in the tRNA corresponding to higher codon-weighted frequency. We found that elongation latency overall led to slightly slower elongation with increasing iterations (Figure S10C, N=3000 runs for each iteration data point), highlighting that simple optimization of relative abundance towards more monotonic features does not necessarily lead to faster elongation latencies but also that elongation speed is relatively robust to small changes in monotonicity.

Figure S9. Most monotonic perturbations away from the CD-CAD-discovered non-monotonic fast tRNA distribution lead to slower elongation latencies. **A.** Pairwise tRNA perturbations across nearest-neighbor tRNA ordered by corresponding codon frequency. In each of the 39 pairs the more weighted tRNA is set to be 10% more abundant than the less weighted tRNA, i.e., towards a more monotonic distribution. Top: An example of a single pairwise tRNA perturbation in which relative abundance is moved from Glu2 to Val1 such that the relative abundance of Val1 is 10% higher than Glu2 and no other tRNA are changed. Both the original and perturbed distributions are shown overlaid (green and blue, respectively). Bottom: Predicted change in average elongation latency relative to baseline for each pairwise tRNA (brown and green color of bars represent whether the change in elongation latency is positive or negative, respectively; average computed from N=30 runs). Perturbation of the relative abundance of Leu1 results in the largest change in elongation latency (marked with an asterisk). **B.** Sequential addition of each of the 39 pairwise tRNA perturbations. Top: The final 39-pairwise tRNA perturbed distribution (blue) overlaid on the original CD-CAD discovered tRNA distribution (green). Bottom: Predicted change in average elongation latency relative to baseline for each additional pairwise perturbation (e.g., bar at Gly3 corresponds to the elongation latency of the CD-CAD-found distribution with one pairwise perturbation and the bar at Arg5 is the elongation latency of the CD-CAD-found distribution with 39 sequential pairwise perturbations; average computed from N=3000 runs).

Figure S10. Iteratively increasing the monotonicity of the CD-CAD-discovered non-monotonic fast tRNA distribution leads to slower elongation latencies. **A.** The first and final iterations of 39-sequential pairwise tRNA perturbations, with tRNA ordered by relative abundance of corresponding codons (perturbed distribution in blue, baseline CD-CAD discovered distribution in green). **B.** Each additional 39-sequential pairwise tRNA perturbation iteration leads to higher monotonicity of tRNA distributions, quantified as the percent of pairwise tRNA neighbors in which the tRNA corresponding to higher codon-frequency has greater relative abundance. **C.** Average elongation latency is slower compared to baseline elongation latency at each iteration and overall becomes slower with increasing monotonicity (each elongation latency iteration data point computed from N=3000 runs).

We further took the reviewer's advice to decompose the non-monotonicity of our distributions via an expression-stratified analysis (Figure S11). We found that, indeed, low expressing genes and high expressing genes prioritize relative tRNA abundance in different ways, consistent with low and high expressing genes having different corresponding codon prioritization. For example, the highest expressing genes (top 10%), intermediate expressing genes (middle 80%), and lowest expressing genes (bottom 10%) all led to different ranked orders of tRNA weighted by corresponding codon frequency (Figure S11B, C, D left column). Reordering these subpopulation distributions to match the order of tRNA in the baseline analysis (all genes) revealed distinct non-monotonic 'component' distributions that underlie the non-monotonicity of the overall distribution (Figure S11A and S11B, C, D right

column). For example, while Ala1B has lower relative abundance in low expressing subpopulations, higher abundance of Ala1B in intermediate and high expressing subpopulation leads to the relatively higher abundance in the overall gene analysis. As another example, Val1 is more prioritized in the high expressing subpopulation compared to the intermediate and low expressing subpopulations likely driving the higher relative abundance of Val1 in the overall population.

Figure S11. Optimizing tRNA distributions for fast translation of gene subpopulations stratified by expression level reveals distinct component distributions underlying the non-monotonic overall tRNA relative abundance distribution. **A.** Baseline CD-CAD-optimized relative abundance of tRNA for fast expression across all genes. **B, C, D.** CD-CAD optimization for fast tRNA distributions across high expression, intermediate expression, and low expression gene subsets. Left: tRNA distributions rank-ordered by codon-usage within gene subsets. Right: tRNA distributions ordered in the same way as the baseline tRNA distribution.

To take our gene stratification analysis to the limit and understand optimization at the single gene level we next analyzed the most highly expressed gene, *ompF* (Figure S12). We found that, even at a single gene level, tRNA abundance distributions are optimally non-monotonic when ordered by codon usage across all genes (Figure S12A) and also when ordered by codon usage for the one gene (Figure S12B, C), though to a lesser extent. Pairwise perturbations and iterative generation of a monotonic distribution resulted in slower average elongation latencies, consistent with the CD-CAD-produced distribution being one local optima (Figure S12, D, E, F, G, H). As with larger gene sets, uneven splitting of relative abundance across isoform tRNA is more optimal for overall speed. Intuitively, elongation latency is also less sensitive to variance in tRNA associated with less-used codons: for example, while Glu2 has a high relative abundance, pairwise perturbation of Glu2 and Ile1 that significantly lowers Glu2 abundance has a relatively small impact on elongation latency (Figure S12D, E). Because Glu2 had an unexpectedly high abundance, we also tried moving mass from Glu2 to the more codon-abundant Val1 which had unexpectedly lower abundance (i.e., 4% absolute decrease of Glu2 and 4% absolute increase in Val1). We found that elongation latency indeed became faster (0.6% faster elongation latency) indicating that a slightly faster optima is possible and that our evolutionary algorithm may have found a local optima in this singular case.

Finally, we analyzed the non-monotonicity of the RED-20 sfGFP tRNA distribution (CD-CAD optimized and experimentally tested, Figure 5; Figure S13A). In this RED-20 case, there is no degeneracy in translation: a single tRNA encodes a single codon, meaning that the tRNA distribution should be able to perform at optimal speed even with a monotonic distribution. Close analysis of relative codon usage and pairwise perturbations supported this hypothesis (Figure S13 B, C, D). For example, AAA and CUA have identical codon usage in RED-20 sfGFP; as such, although Lys and Leu3 (the corresponding tRNA) appear differently ordered producing non-monotonicity, they are swappable and small variations in their relative abundances have approximately no impact on elongation latency. As another example, Gly2 encoding GCA is only slightly more relatively abundant than AAA and CUA; as such, unsurprisingly, elongation latency is relatively insensitive to moving some relative abundance from Gly2 to Leu3. Overall, pairwise perturbations and shifting the CD-CAD distribution to be fully monotonic led to relatively negligible changes in total elongation latency, highlighting that the CD-CAD predicted tRNA distribution for RED-20 sfGFP has a broad range of similar optima, some of which are monotonic and some of which are non-monotonic (Figure S13D, E, F, G).

Figure S12. CAD-optimization and sensitivity analysis of the single high expressing gene *ompF* reveals non-monotonicity intrinsic to degeneracy of translation. **A.** Relative tRNA abundance produced by CD-CAD optimized for fast expression of *ompF*, with tRNA ordered by overall transcriptome codon abundance. **B.** The same relative tRNA abundance reordered by codon usage in *ompF*. **C.** Codon usage in *ompF*. **D.** An example of pairwise perturbation of tRNA relative abundance (perturbed distribution in blue, baseline CD-CAD discovered distribution in green). **E.** Changes in average elongation latency resulting from pairwise perturbation across every neighboring pair of tRNA ordered by codon usage in *ompF* (average computed from N=3000 runs of each perturbation). **F.** Fully monotonic tRNA distribution (blue) resulting from iteratively making the original CD-CAD discovered tRNA distribution (green) more monotonic. **G.** Monotonicity increasing over iterations (quantified

as the percent of pairwise tRNA neighbors in which the tRNA corresponding to higher codon frequency has greater relative abundance). **H.** Predicted average elongation latency is slower compared to baseline with increasing monotonicity (each elongation latency iteration data point computed from N=3000 runs).

Figure S13. CAD optimization and sensitivity analysis of RED20-encoded GFP in a minimal tRNA system reveals how expression in a non-degenerate translation system can behave optimally even with a monotonic tRNA abundance distribution. A. Relative tRNA abundance produced by CD-CAD optimized for fast expression of RED-20 GFP in a minimal tRNA system. **B.** Codon usage in RED20-encoded GFP. **C.** An example of pairwise perturbation of tRNA abundance (perturbed distribution in blue, baseline CD-CAD discovered distribution in green). **D.** Changes in elongation latency resulting from pairwise perturbation across every neighboring pair of tRNA ordered by codon usage in RED20-encoded GFP (average computed from N=3000 runs of each perturbation). **E.** Fully monotonic tRNA distribution (blue) resulting from iteratively making the original CD-CAD discovered tRNA distribution (green) more monotonic. **F.** Monotonicity increasing over iterations (quantified as the percent of pairwise tRNA neighbors in which the tRNA

corresponding to higher codon frequency has greater relative abundance). **G.** Predicted average elongation latency is only marginally slower overall compared to baseline with increasing monotonicity (each elongation latency iteration data point computed from N=3000 runs).

Page 9: No estimated elongation rates from the experiments are given. Why not? Would it not be possible to calculate this given the system is controlled to properly assess the accuracy of the model?

While our modeling framework produces unfitted absolute estimates of translation elongation latency these values are estimated via a model developed for *E. coli* cytoplasm using native (i.e., modified) tRNA. To make direct absolute rate comparisons in our PURE-based experimental system we would need to observe the detailed kinetics of each step of translation elongation using synthetic RNA as tRNA surrogates (a project well beyond the scope of the work reported here). Instead, leveraging the fact that most elongation latency is due to physical transport of tRNA, we can confidently predict and observe relative quantitative differences in elongation rates.

Page 9: A comparison of several different tRNA distributions is shown for a single fluorescent reporter. This is a rather limited experiment and would benefit from seeing whether similar predictive power is seen for other types of protein. For example, considering genes where biases in amino acids are seen would offer a chance to explore some of the questions that remain around the encoding of genes and the effects these have on translational dynamics.

With thanks. We agree fully. Many interesting scientific and technical questions can be explored and answered via the platform we have developed and report on here for the first time. We are confident in the reported predictions and observations. We look forward to us and others building on this initial work in future research. We have added text to the Discussion that better highlights what questions and work might best be pursued.

Page 10: Why is there such large variability between batches? It seems like the expended trends are observed, but it is a little too strong to say they “matched” the design.

The functional quality of our synthetic RNA tRNA surrogates degrades over time (even when stored at -80C) and with freeze/thaw cycles; as we ran more experiments, we saw that our synthetic tRNAs supported lower and lower expression levels. We note this observation in the Discussion. We recommend that any future experiments using synthetic tRNAs be performed immediately upon receiving those tRNAs to minimize variance between batches arising due to tRNA degradation.

In spite of this batch-to-batch decline in activity, we were excited to observe that synFast and synSlow are faster and slower than each other, across all batches. Not only that, but the relative difference in the translation rates of synFast and synSlow are similar to what we

predicted, especially when averaging across experiments (see Figure S8). We agree with the reviewer that it is perhaps too strong to say these experiments exactly “matched” the as-designed predictions in an absolute quantitative sense and have updated the language to reflect the clear qualitative agreement.

Page 11: I struggled to fully understand how you leveraged previous simulation data in this work. It would have been far more helpful to include the source code of the simulations and extended model to be able to assess precisely what has been done. Why was this not included?

We again apologize for the oversight of not including a link to all code used in our modeling and simulations. A Github repository containing all code has been linked in the manuscript and here: <https://github.com/EndyLab/tRNACAD>

We have also clarified our description of how we use previous simulation data in the Main Text and Methods.

Figure 1A: Scale bar is not labeled, and text is impossible to read on the x-axis. It would be helpful to ensure all text is of a similar and readable size (e.g., 8pt). This comment relates to all the figures, as text size in many of them is a problem that needs to be fixed.

We included a label (‘Frequency (%)’) for scale bars across all figures and improved text size and readability across all figures.

Reviewer #2 (Remarks to the Author):

the manuscript addresses an interesting element of synthetic biology considering engineering of synthetic cells specifically the required abundance of different tRNAs. The manuscript reports on the generation of a genetic algorithm to do this. The manuscript is clearly written and combines the development of the computational model with in vitro validation studies.

I have the following comments:

1. The manuscript had two conditions on tRNA abundance, I understand the need for the first one, but I wonder if it could be explained why the overall tRNA concentration was constant.

We thank the reviewer for their important question. We had three reasons for holding total tRNA abundance constant:

- (1) We wanted to focus on understanding specifically how changes in relative tRNA abundances impact translation rates. Allowing total tRNA concentrations to be a free parameter would increase the search space to an extent that would make it difficult to isolate the effects of changing total abundances versus relative abundance (e.g., in the limit, a system with zero tRNA would take infinite time to translate).
- (2) We expect that, on average, increasing tRNA concentrations will lead to faster elongation rates, up until occupied volume fraction results in increases in effective viscosity so great that hindering molecular transport reduces overall elongation rates (Figure 6, Maheshwari et al., mBio, 2022). Also, when building synthetic cellular systems, there will ultimately be a cap on how many tRNA can be included or natively produced during propagation given the tradeoff of needing other molecules. As an orthogonal means for optimizing cell performance without requiring additional resources we choose to explore here only changing the relative composition of a fixed amount of total tRNA mass.
- (3) Practically, we discovered that if we kept total tRNA concentration fixed we could repurpose previous colloidal physics translation voxel simulations (performed with the same fixed total tRNA concentration, and which took ~300,000 CPU hours to produce (Maheshwari et al., 2022)). By using this already “trained” model we could realize a ~100,000-fold improvement in simulation time and cost.

The Main Text has been modified to offer these reasons to the reader.

2. In the first part we see the the tRNA abundance based on codon frequency performs better than wild type. The CD-CAD approach alter this and has higher frequencies of the most widely used tRNAs. How much faster (better) are these abundances proposed by CD-CAD than using the abundances that correlate with codon usage? i.e. what is the benefit of CD-CAD over this much simpler optimisation? Is the cost worth the improvement?

Thank you for asking for this clarification! We show that for the wild-type *E. coli* transcriptome, the CD-CAD optimized tRNA distribution results in elongation latency of 175 ms (Figure 3D) while the codon-usage correlated results in elongation latency of 185 ms

(Figure 2D). That is, the CD-CAD optimized distribution performs ~5% better than the codon-usage correlated distribution.

We performed additional simulations to compute codon-usage correlated elongation latencies for the codon-reduced transcriptome and RED20-GFP, as below (Figure S14). For the codon-reduced transcriptome, we found that the codon-weighted tRNA distribution results in an elongation latency of 163 ms vs. 157 ms for the CAD-optimized tRNA distribution. That is, the CD-CAD distribution performs ~4% better than the codon-usage correlated distribution (Figure S14A). For RED20-encoded GFP, we found that the codon-weighted tRNA distribution results in an elongation latency of 163 ms vs. 160 ms for the CAD-optimized tRNA distribution. That is, the CD-CAD distribution performs ~2% better than the codon-usage correlated distribution. The very small improvement for RED20-encoded GFP performance is consistent with our result that monotonic distributions can be as effective as CAD-encoded distributions for single genes in non-degenerate genetic codes (Figure S13), as described in our response to Reviewer 1.

Regarding whether the cost is worth the improvement, we start by noting that for any synthetic cellular system, *some* tRNA distribution must be specified *a priori*. Previous work has either used a uniform tRNA distribution or wild type distribution (Calles et al., 2019; Shimizu et al., 2005). Stated differently, nobody has as-yet implemented a codon-weighted tRNA distribution experimentally. Our computational tool coupled with TINA enables straightforward design and testing of rational distributions (e.g., monotonic increasing or codon-weighted). Using CAD for optimization takes ~1.5 hrs, which compared to the typical time needed for experimental design and implementation is negligible. Ultimately, whether a CAD-optimized tRNA distribution or codon-usage correlated distribution should be used will depend on the particularities of any given use case. We anticipate that for design of complex, multi-gene synthetic cellular systems, every bit of performance matters, especially performance that comes without the biological cost of additional cell mass being used towards tRNA (i.e., when optimizing tRNA distributions more total tRNA does not need to be produced, rather the same total tRNA only needs to be distributed in different ways across tRNA species).

We have added these ideas to the Discussion.

Figure S14. CAD-produced fast tRNA distributions perform better than codon-weighted distributions for both the RED20 codon-reduced transcriptome and RED20-encoded GFP. A. Left: Codon-weighted distribution of the reduced set of tRNA for expression of the RED20 codon-reduced transcriptome. Right: Per-transcript elongation latency distribution for the CAD-produced fast tRNA distribution (green) vs. codon-weighted (purple). **B.** Left: Codon-weighted distribution of the reduced set of tRNA for expression of RED20-encoded GFP. Right: Elongation latency for the CAD-produced fast tRNA distribution (green) vs. codon-weighted (purple).

3. Figure 3 it would be good to avoid the combination of red and green in this figure.

Thank you. We have changed the color scheme to red and blue in this figure.

4. Do we always want to optimise the rate of protein synthesis? Changes in protein folding, (although often through codon optimisation) have shown that they can affect the folding of proteins and alter their function? Is there any consideration for this? i.e. would it always be good to speed up the rate of translation? There is some thought on this in the discussion but it could be expanded.

We take the reviewer's point; we do not always want to maximize the rate of protein synthesis. We recognize other design considerations exist (e.g., co-translational protein folding that requires slow translation across difficult to fold regions) even if the end goal were simply to produce the most amount of a target protein as fast as possible.

Taking a step back, we wished to demonstrate proof of principle for a "design build work" framework for engineering synthetic biological systems. Ultimately, we want this strategy to apply broadly to many systems, not only translation. We wish for practitioners to be able to specify performance characteristics of their synthetic biological systems, not just maximize or minimize those characteristics. We chose to maximize and minimize translation rates to ground our first-ever exploration in a particular system via a simply accessible performance

specification (fast or slow), but hope that such an approach will be used across systems and for more esoteric specifications.

In response to this question, we've added text to the Discussion that clarifies our motivation for maximizing and minimizing translation rate and expands on specifying performance characteristics more generally.

Reviewers' Comments:

Reviewer #1:

Remarks to the Author:

I thank the authors for their thoughtful consideration of my comments and the extensive revision they have presented which addresses all of my concerns and, I believe, helps strengthen the work. This will make an excellent addition to the literature.

Reviewer #2:

Remarks to the Author:

The revision addresses my original comments.

REVIEWERS' COMMENTS

Reviewer #1 (Remarks to the Author):

I thank the authors for their thoughtful consideration of my comments and the extensive revision they have presented which addresses all of my concerns and, I believe, helps strengthen the work. This will make an excellent addition to the literature.

We thank the reviewer.

Reviewer #2 (Remarks to the Author):

The revision addresses my original comments.

We again thank the second reviewer.